# Effect of Peptide Receptor Radionuclide Therapy in Combination with Temozolomide against Tumor Angiogenesis in a Glioblastoma Model

**DOI:** 10.3390/cancers13195029

**Published:** 2021-10-08

**Authors:** Sang Hee Lee, Ji Young Choi, Jae Ho Jung, In Ho Song, Hyun Soo Park, Nunzio Denora, Francesco Leonetti, Sang Eun Kim, Byung Chul Lee

**Affiliations:** 1Department of Nuclear Medicine, Seoul National University Bundang Hospital, Seoul National University College of Medicine, Seongnam 13620, Korea; kkpling@snu.ac.kr (S.H.L.); cjy0929@snu.ac.kr (J.Y.C.); jjh@bioimaging.co.kr (J.H.J.); 99269@snubh.org (I.H.S.); hyuns@snu.ac.kr (H.S.P.); 2Department of Transdisciplinary Studies, Graduate School of Convergence Science and Technology, Seoul National University, Seoul 08826, Korea; 3Department of Pharmacy—Pharmaceutical Sciences, University of Bari “A. Moro”, 70121 Bari, Italy; nunzio.denora@uniba.it (N.D.); francesco.leonetti@uniba.it (F.L.); 4Center for Nanomolecular Imaging and Innovative Drug Development, Advanced Institute of Convergence Technology, Suwon 16229, Korea; 5Department of Molecular Medicine and Biopharmaceutical Sciences, Graduate School of Convergence Science and Technology, Seoul National University, Seoul 08826, Korea

**Keywords:** tumor angiogenesis, temozolomide, combination therapy, peptide receptor radionuclide therapy

## Abstract

**Simple Summary:**

Glioblastoma multiforme (GBM) is an aggressive brain tumor characterized by intense angiogenesis. Thus, tumor angiogenesis-related receptors, such as the cell adhesion molecule integrin α_v_β_3_, are potential biomarkers for cancer diagnosis and therapy. In this study, we aimed to investigate the therapeutic efficacy of peptide receptor radionuclide therapy (PRRT) with ^188^Re-IDA-D-[c(RGDfK)]_2_ (11.1 MBq). Our results revealed that PRRT combined with temozolomide markedly reduced the tumor volume compared with monotherapy. In summary, ^188^Re-IDA-D-[c(RGDfK)]_2_ might be an effective radiotherapeutic agent for the treatment of GBM.

**Abstract:**

Cell adhesion receptor integrin α_v_β_3_ is a promising biomarker for developing tumor-angiogenesis targeted theranostics. In this study, we aimed to examine the therapeutic potential of peptide receptor radionuclide therapy (PRRT) with ^188^Re-IDA-D-[c(RGDfK)]_2_ (11.1 MBq). The results showed that the tumor volume was significantly decreased by 81% compared with the vehicle-treated group in U87-MG xenografts. The quantitative in vivo anti-angiogenic responses of PRRT were obtained using ^99m^Tc-IDA-D-[c(RGDfK)]_2_ SPECT and corresponded to the measured tumor volume. PRRT combined with temozolomide (TMZ) resulted in a 93% reduction in tumor volume, which was markedly greater than that of each agent used individually. In addition, histopathological characterization showed that PRRT combined with TMZ was superior to PRRT or TMZ alone, even when TMZ was used at half dose. Overall, our results indicated that integrin-targeted PRRT and TMZ combined therapy might be a new medical tool for the effective treatment of glioblastoma.

## 1. Introduction

Angiogenesis is essential for tumor growth and metastasis [1]. Tumor angiogenesis-related receptors are promising biomarkers for cancer diagnosis and therapy [2]. The cell adhesion molecule integrin α_v_β_3_ is a specific marker of tumor angiogenesis and plays a crucial role in the advancement and metastatic spread of cancer [3]. Therefore, antagonists against integrin α_v_β_3_ were designed and evaluated either for tumor-specific anticancer therapy or combined with various therapeutic anticancer agents [4]. In addition, integrin α_v_β_3_ can be used to assess expression status in vivo noninvasively. Thus, it may be valuable for evaluating the efficacy of anti-integrin treatment in reducing tumor growth and spread in order to improve therapy planning and monitoring of anti-angiogenic therapies [5,6]. Tripeptide Arg-Gly-Asp (RGD) sequence has been proven effective as a specific binding motif for integrin receptors, and since, numerous radionuclide-labeled RGD peptides targeting integrin α_v_β_3_ have been developed and used in positron emission tomography (PET) and single-photon emission computed tomography (SPECT), which can longitudinally diagnose tumor angiogenesis in cancer [7,8,9,10,11,12,13,14,15,16,17]. Theranostics combines diagnostic imaging and therapy into a single platform and is considered the next generation of personalized medicine [18]. Nuclear medicine imaging, along with radiotherapeutic agents, is effective in planning and monitoring biology-driven personalized radiotherapy. For instance, ^99m^Tc/^188^Re is one of the most promising pairs owing to their favorable nuclear properties for diagnostic imaging (t_1/2_ = 6 h, gamma energy of 141 keV) and tumor radiotherapy (t_1/2_ = 17 h, maximum beta energy of 2.12 MeV), respectively. Moreover, ^99m^Tc and ^188^Re can be easily obtained by periodic aseptic elution of ^99^Mo/^99m^Tc- and ^188^W/^188^Re-generator, respectively, and are thus suitable for routine clinical use.

Glioblastoma multiforme (GBM) is a highly vascularized cancer [19]. Glioma cells produce proangiogenic factors, including vascular endothelial growth factor (VEGF); additionally, high levels of these factors are correlated with high-grade malignancy and poor prognosis [20,21]. Temozolomide (TMZ), which is spontaneously cleaved in vivo and generates the reactive DNA alkylating agent monomethyl triazenoimidazole carboxamide that promotes apoptosis, is used as the current standard chemotherapeutic agent for newly diagnosed GBM. It is typically used for the treatment of GBM in conjunction with radiation therapy.

We have previously reported the development of ^99m^Tc- and ^188^Re-labeled RGD dimer peptides (^99m^Tc- and ^188^Re-IDA-D-[c(RGDfK)]_2_), including the ^99m^Tc- or ^188^Re(CO)_3_-(iminodiacetate, IDA) core for tumor angiogenesis imaging and radiotherapy [18]. Both radiolabeled RGD peptides have similar activity: i) very high integrin-binding affinity (0.4–0.5 nM) and ii) high tumor accumulation but relatively low liver and intestinal uptake. ^99m^Tc-IDA-D-[c(RGDfK)]_2_ SPECT not only showed remarkable integrin-targeting specificity, high sensitivity, and desirable excretion kinetics in tumor xenografts, but was also an efficacious and safe radiotracer for diagnosing integrin α_v_β_3_-expressing tumors [22,23].

In this study, we evaluated the possible use of peptide receptor radionuclide therapy (PRRT) of ^188^Re-IDA-D-[c(RGDfK)]_2_ with ^99m^Tc-IDA-D-[c(RGDfK)]_2_ as a promising theranostic strategy in U87-MG human glioblastoma xenografts. Furthermore, as the main aim of this study, we examine the possible synergistic effect of PRRT and TMZ and whether this combination is more effective than individual compounds.

## 2. Materials and Methods

### 2.1. Chemicals

Reagents and solvents were commercially purchased from Merck (Seoul, Korea) and used without further purification unless otherwise specified. The precursors (IDA-D-[c(RGDfK)]_2_ and IDA-D-[c(RADfK)]_2_) and three radiotracers (^99m^Tc-IDA-D-[c(RGDfK)]_2_, ^188^Re-IDA-D-[c(RGDfK)]_2_, and ^188^Re-IDA-D-[c(RADfK)]_2_), and Q-dot 605-labeled RGD peptide (Q-dot 605-D-[c(RADfK)]_2_) were prepared according to previously described methods [18,24]. The ^99^Mo/^99m^Tc- and ^188^W/^188^Re-generator was purchased from Samyoung Unitech (Seoul, Korea) and Enviro Korea (Daejeon, Korea), respectively.

### 2.2. Cell Culture and Treatments

The integrin α_v_β_3_ positive human glioblastoma cell, Uppsala87-Malignant Glioma (U87-MG, American Type Culture Collection (ATCC)^®^ HTB-14^TM^; ATCC-LGC Standard, Wesel, Germany) was cultured in Dulbecco’s Modified Eagle Medium (DMEM, Gibco, Grand Island, NY, USA) supplemented with 10% fetal bovine serum (FBS) and 1% antibiotics/antimycotics (Gibco, Grand Island, NY, USA) at 37 °C in 5% CO_2_.

### 2.3. Preparation of Tumor-Bearing Mice

A suspension of human glioblastoma U87-MG cells was prepared (5 × 10^6^ cellsmL^−1^). A total of 0.1 mL of cell suspension was injected subcutaneously into the right flank of a 6–7-week-old male BALB/c nu/nu nude mice (20–25 g, Orient Bio Inc., Seongnam, Korea). In the case of pharmacokinetic studies of ^99m^Tc-IDA-D-[c(RGDfK)]_2_, we used tumor xenograft mice whose tumor cells were inoculated in the right shoulder and performed the SPECT imaging study when tumor volume reached 68.7 ± 18.8 mm^3^ at 10 days after injection of U87-MG cells. Evaluations of integrin-targeted blocking and radiotherapy treatment on the growth of U87-MG xenografts were carried out when tumor volume reached 59.4 ± 14.9 mm^3^ (For authentic “cold” Re-peptide (^185/187^Re-IDA-D-[c(RGDfK)]_2_) treatment) and 66.2 ± 15.3 mm^3^ (For radiotherapy (^188^Re-IDA-D-[c(RGDfK)]_2_) treatment) at 10 days after injection of U87-MG cells. Experiments with vehicle, negative control peptide (^188^Re-IDA-D-[c(RADfK)]_2_) or ^188^Re-IDA-D-[c(RGDfK)]_2_ treatments (11.1 MBq) in U87-MG xenografts were carried out when tumor volume reached 40.3 ± 14.8 mm^3^ at 9 days after injection of U87-MG cells. The combination therapy of TMZ with ^188^Re-IDA-D-[c(RGDfK)]_2_, including single doses of TMZ, was carried out when tumor volume reached 47.9 ± 8.6 mm^3^ at 9 days after injection of U87-MG cells. The body weight of mice for each group was maintained during all therapy experiments. All animal studies were performed under the protocols approved by the Institution Guidelines on the Use and Care of Animals. Mouse protocols were approved by Committee of Seoul National University Bundang Hospital (IACUC No. BA1211-117/080-01, approved on 27 November 2012). Animal Care and Use Committee. We performed the caliper measurements of the longest perpendicular tumor diameters.

### 2.4. Radiotherapy for the U87-MG Xenografts

All mice were intravenously injected with saline (Control group), “cold” ^185/187^Re-coordinated peptide (^185/187^Re-IDA-D-[c(RGDfK)]_2_: 0.013, 5, and 10 mgkg^−1^) or ^188^Re-labeled peptide (^188^Re-IDA-D-[c(RGDfK)]_2_: 3.7, 7.4, 11.1, and 18.5 MBq) every 4 days over a period of 14 days, and primary tumor growth was monitored daily for 2 weeks. Tumor volume was based on caliper or CT measurements. In cold, ^185/187^Re-IDA-D-[c(RGDfK)]_2_ treatments, the dose of the ligand (0.013 mgkg^−1^) was estimated from the molar activity of ^188^Re-IDA-D-[c(RGDfK)]_2_ in an 18.5 MBq radiotherapy dose.

### 2.5. Combination Therapy of TMZ with ^188^Re-IDA-D-[c(RGDfK)]_2_ in the U87-MG Xenografts

U87-MG xenografts were assigned to various groups and injected intravenously for 2 weeks with planned treatments: saline (control), TMZ (2 or 5 mgkg^−1^), ^188^Re-IDA-D-[c(RGDfK)]_2_ (11.1 MBq), or ^188^Re-IDA-D-[c(RGDfK)]_2_ (11.1 MBq) + TMZ (2 mgkg^−1^). TMZ was administered once a week for 14 days. ^188^Re-IDA-D-[c(RGDfK)]_2_ was administered in 4 injections every 4 days over a period of 14 days. The experimental groups (4 mice for each) and corresponding control groups (4 mice) were examined. At the end of the experiment, animals were sacrificed, and tumors were excised and weighed.

### 2.6. Immunohistochemistry

Tumor tissues were harvested on day 14 after the treatment and immediately fixed in 10% formalin solution. Frozen tissue sections (5 μm) were placed onto glass slides. The tumor sections were stained with anti-integrin α_v_β_3_ rabbit antibody (1:1000, Millipore) at 4 °C for 16 h. The sections were then incubated with anti-rabbit secondary antibody (biotinylated) at room temperature for 1 h. The detection system (Vector Laboratories, Burlingame, CA, USA) was applied according to the manufacturer’s instructions. The nuclei were counter-stained with hematoxylin (Invitrogen, Carlsbad, CA, USA). For assessment of tumor vascularization, mouse anti-human CD31 monoclonal antibody (diluted 1:40, Dako) staining was performed on acetone-fixed cryosections using a blood vessel staining kit (ECM590, Millipore Ireland, Cork, Ireland). For another assessment of integrin α_v_β_3_ receptors in dissected tumor tissues, anti-integrin α_v_β_3_ (SC-7312, Santa Cruz Biotech, Santa Cruz, CA, USA) was used. Primary antibody against γH2AX (diluted 1:50, Abcam) was applied to acetone-fixed cryosections and incubated overnight. Secondary donkey anti-goat antibody (FITC conjugated, Invitrogen, Carlsbad, CA, USA) was applied and incubated for 1 h. Immunohistochemical staining was performed at 30 days after excision of tumor due to the half-life of radioisotope (Re-188). Images were acquired using Axioscope A1 fluorescent microscope (Carl Zeiss, Jena, Germany) or AxioCam MRc5 (Carl Zeiss, Jena, Germany) and analyzed with Axiovision software (version 4.4, Carl Zeiss Meditec, Jena, Germany). The nuclei were counter-stained with hematoxylin (Invitrogen, Carlsbad, CA, USA).

### 2.7. Confocal Microscopy

Fluorescence images of Q-dot 605-D-[c(RGDfK)]_2_ were collected using a Zeiss LSM510 META Confocal Imaging System with a Chameleon laser system (Carl Zeiss, Jena, Germany). All images were taken with an EC-Plan Neo-Fluar 40× (NA 1.3) oil immersion lens. The Q-dot 605 was excited at 543 nm, and emission was monitored from 590 to 620 nm. Images were analyzed using Zeiss LSM software (Carl Zeiss, Jena, Germany).

### 2.8. Tumor Growth Measurement

For caliper measurements, tumor length (longitudinal diameter) and tumor width (transverse diameter) were measured, and the tumor volume was calculated according to the following formula: tumor volume = (length × width^2^)/2.

Animal CT imaging was performed using NanoSPECT/CT (Bioscan Inc., Washington, DC, USA) consisting of a low-energy X-ray tube and a precision-motion translation stage. A total of 180 projections were acquired with the X-ray source set at 45 kVp and 177 mA. Two-dimensional slices were reconstructed using an Exact Cone Beam Filter Back Projection algorithm with a Shepp–Logan filter. CT images were reconstructed on a voxel/pixel size of 0.20:0.192 mm, providing image sizes (x, y, z) of 176 × 176 × 136 with an image resolution of 48 mm.

### 2.9. SPECT Image Analysis

Animal SPECT/CT imaging was acquired using a NanoSPECT/CT using low-energy and high-resolution pyramid collimator. Mice were placed in a prone position on the bed and kept under anesthesia with 2% isoflurane. SPECT images were obtained at 0 to 180 min after intravenous injection of ^99m^Tc-IDA-D-[c(RGDfK)]_2_ (18.5 MBq, *n* = 4). After SPECT imaging, whole-body CT images were obtained in 24 projections over a 10 min period using a 4-head scanner with 4 × 9 (1.4 mm) pinhole collimators in helical scanning mode. Image reconstruction and quantification of micro-SPECT and CT images was performed using the software programs HiSPECT (version 1.0, Bioscan Inc. Washington, DC, USA) and InVivoScope software (version 1.43, Bioscan Inc. Washington, DC, USA), respectively. The percentage of the injected dose per gram of tissue (%IDg^−1^) was determined from the radionuclide uptake in the region of interest (ROI) on the tumor after intravenous injection of ^99m^Tc-IDA-D-[c(RGDfK)]_2_.

### 2.10. Statistical Analysis

Statistical software SPSS version 10.1 (SPSS Inc., Chicago, IL, USA) was used for analyzing results. Multiple group comparisons were made using one-way ANOVA followed by post hoc test (Bonferroni correction). Differences with a *p*-value less than 0.05 were considered as significant. 

## 3. Results

### 3.1. Pharmacokinetic Studies of ^99m^Tc-IDA-D-[c(RGDfK)]_2_ in U87-MG Xenografts

As shown in Figure 1A, ^99m^Tc-IDA-D-[c(RGDfK)]_2_ had prominent tumor accumulation and retention potential with rapid general clearance mainly through the kidneys and to a lesser extent through the liver. The quantified tumor uptake of ^99m^Tc-IDA-D-[c(RGDfK)]_2_ was measured from the ROI of SPECT images and expressed as a percentage of the injected dose per gram tissue (%IDg^−1^) (Figure 1B). 

The noninvasive measurement of tumor size and radionuclide uptake using ^99m^Tc-IDA-D-[c(RGDfK)]_2_ SPECT was compared with conventional measurements of caliper and gamma counter, respectively. Our analysis revealed a positive correlation between in vivo animal SPECT/CT semi-quantification image analysis and ex vivo tumor radionuclide uptake measurements (R^2^ = 0.894; Appendix A). Thus, ROI-derived %IDg^−1^ values provided high-confidence values for assessing the angiogenic response in tumor-bearing mice (*n* = 13) with tumor volumes of 64.2–3569.4 mm^3^.

To anticipate the therapeutic efficacy and side effects of ^188^Re-IDA-D-[c(RGDfK)]_2_, we assessed the pharmacokinetic (PK) parameters of ^99m^Tc-IDA-D-[c(RGDfK)]_2_ in three tumor-bearing nude mice. PK parameters were derived using nonlinear regression curve fitting. The area under the curve (AUC_0-__∞_) that was obtained by plotting concentration versus time for ^99m^Tc-IDA-D-[c(RGDfK)]_2_ in the liver and muscle was proportionally 10- and 200-fold lower than that in the tumor, respectively. The AUC_0_-∞ of tumor was slightly higher than that of the kidneys. The maximum concentrations (C_max_) of ^99m^Tc-IDA-D-[c(RGDfK)]_2_ in the tumor and kidneys were 13.3 and 24.0% IDg^−1^, respectively.

The elimination half-life (T_1/2 elim_) of ^99m^Tc-IDA-D-[c(RGDfK)]_2_ in the liver was 83.39 min, which was 1.10-fold higher than that in the tumor. The clearance rate (Cl) and T_1/2 elim_ of ^99m^Tc-IDA-D-[c(RGDfK)]_2_ in the kidneys were 0.053 mLmin^−1^ and 28.54 min, respectively. These results partially showed that the kidneys eliminated the radionuclide uptake more rapidly than the tumor. However, these PK parameters from SPECT imaging indicated that the repeated dosing and renal toxicity of ^188^Re-IDA-D-[c(RGDfK)]_2_ need to be considered before pharmacological evaluation.

### 3.2. Anti-Angiogenic Effect of ^188^Re-IDA-D-[c(RGDfK)]_2_

For a preliminary study (Appendix A), a single dose (22.2 MBq in 200 μL of saline) of ^188^Re-IDA-D-[c(RGDfK)]_2_ and only saline (200 μL) were injected into U87-MG xenografts (tumor volume = 300 mm^3^). At 3 d post-injection, the tumor volumes of ^188^Re-IDA-D-[c(RGDfK)]_2_-treated mice slightly increased to 350 mm^3^, whereas those of saline-treated mice more than doubled. In addition, the observed %IDg^−1^ of ^99m^Tc-IDA-D-[c(RGDfK)]_2_ (approximately 10% IDg^−1^) of a tumor before ^188^Re-IDA-D-[c(RGDfK)]_2_ injection significantly reduced by 50%. CD31 immunostaining for microvessels, and fluorescence imaging of Q-dot 605-D-[c(RGDfK)]_2_ for integrin receptors also revealed the anti-angiogenic efficacy of ^188^Re-IDA-D-[c(RGDfK)]_2_ on the tumor tissue. The anti-angiogenic effect of integrin-targeted therapy is probably brief since rapid revascularization of tumors occurs after discontinuing anti-VEGF treatment [25,26]. Therefore, injection of ^188^Re-IDA-D-[c(RGDfK)]_2_ was performed every 4 d throughout 14 d based on the preliminary PK parameters of ^99m^Tc-IDA-D-[c(RGDfK)]_2_ and the periodic elution time of the ^188^W/^188^Re-generator.

Before radiotherapy, U87-MG xenografts were treated with saline (control) and ^185/187^Re-coordinated IDA-D-[c(RGDfK)]_2_ (0.013–10 mgkg^−1^ in saline) to clarify whether the cold Re-peptide also has an anti-angiogenic effect as the integrin antagonist (Figure 2A). The results indicated that the amount (0.013 mgkg^−1^) of ^185/187^Re-coordinated IDA-D-[c(RGDfK)]_2_, which was calculated from the molar activity of ^188^Re-IDA-D-[c(RGDfK)]_2_, had no pharmacodynamic effect in the tumor model. Furthermore, when tumors were treated with cold Re-peptide (^185/187^Re-IDA-D-[c(RGDfK)]_2_), the inhibitions of tumor growth showed a linear relationship with doses. Although treatments with ^185/187^Re-IDA-D-[c(RGDfK)]_2_ had a minimal impact on tumors, the PRRT of ^188^Re-IDA-D-[c(RGDfK)]_2_ markedly affected tumor growth (Figure 2C). After 14 d of treatment, ^188^Re-IDA-D-[c(RGDfK)]_2_ at low doses of 3.7 and 7.4 MBq could significantly suppress tumor growth by 33% and 56% (*p* < 0.05), respectively, compared with the control group. In contrast, at higher doses of 11.1 and 18.5 MBq, it could completely inhibit tumor growth (64% and 69% in tumor size reduction, respectively, compared with the control). Overall, ^188^Re-IDA-D-[c(RGDfK)]_2_ treatments showed good tolerance in all experiments and did not result in body weight changes, except for the 18.5 MBq dose.

Immunohistochemical analysis of the dissected tumor tissues with the anti-human CD31 monoclonal antibody and anti-integrin α_v_β_3_ antibody showed reduced microvessel density and anti-angiogenic effects in tumor tissues (Figure 2D,E). These results indicated that integrin-targeted PRRT was superior to integrin-targeted treatment inhibition. CD31 immunostaining data supported the significant anti-angiogenic effect of ^188^Re-IDA-D-[c(RGDfK)]_2_ with increasing radiotherapy doses [27]. The % area of CD31 positive microvessels was decreased to 31% at 5 mgkg^−1^ and 26% at 10 mgkg^−1^ in the ^185/187^Re-IDA-D-[c(RGDfK)]_2_-treated group. In contrast, the ^188^Re-IDA-D-[c(RGDfK)]_2_-treated group showed markedly increased destruction of tumor tissue microvessels in a linear radioactivity dose-dependent manner. Moreover, immunohistochemical staining with anti-integrin α_v_β_3_ antibody revealed the suppression of tumor growth in groups treated with 11.1 and 18.5 MBq ^188^Re-IDA-D-[c(RGDfK)]_2_ accompanied by an 82% and 92% decrease in integrin expression levels, respectively, compared with the control.

### 3.3. Selectivity of Radiotherapy

After determining the optimal effective radiotherapeutic dose at 11.1 MBq, we investigated whether ^188^Re-IDA-D-[c(RGDfK)]_2_ has a selective anti-angiogenic effect on U87-MG tumors compared with the negative control peptide ^188^Re-IDA-D-[c(RADfK)]_2_, which does not bind integrin α_v_β_3_ owing to the addition of a single methyl group, changing glycine to alanine [28,29]. We analyzed the ability of ^188^Re-IDA-D-[c(RGDfK)]_2_ in suppressing tumor growth in U87-MG xenografts after 14 d of treatment, and the results were compared with those of “vehicle” (0.013 mgkg^−1^ of ^185/187^Re-IDA-D-[c(RGDfK)]_2_ in saline) and negative control peptide-treated groups (Figure 3). No suppression of tumor growth was observed in mice xenografts when treated with 11.1 MBq of negative control peptide. In contrast, we observed 81% tumor volume reduction in the ^188^Re-IDA-D-[c(RGDfK)]_2_-treated group compared with the vehicle-treated group. 

Based on macroscopic observations, the dissected tumors in the ^188^Re-IDA-D-[c(RGDfK)]_2_-treated group appeared less vascularized than those in the vehicle- and negative control peptide-treated groups (Figure 3C). Histological examination also revealed significant differences among the groups. The ^188^Re-IDA-D-[c(RGDfK)]_2_-treated tumor tissues showed a decrease in microvessel density of approximately 60% as assessed by CD31 immunostaining (Figure 3D) and a significant decrease in positive integrin α_v_β_3_ staining. Therefore, the anti-angiogenic effects of ^188^Re-IDA-D-[c(RGDfK)]_2_ might be partly responsible for the delay in tumor growth. We also found that beta radiation from ^188^Re-IDA-D-[c(RGDfK)]_2_ damaged cancer cells in the tumor angiogenic region and reduced the concentration of integrin α_v_β_3_ receptors. In the nuclei of tumor tissues counter-stained with hematoxylin, ^188^Re-IDA-D-[c(RGDfK)]_2_-treated tumor tissues showed a damaged architecture and less intense staining, leading to massive necrosis. Radiotherapy-induced tumor necrosis was evident in the lower half of the field, whereas several scattered large tumor cells with generative features were observed in the upper half of the field (Figure 3E). The inset picture of ^188^Re-IDA-D-[c(RGDfK)]_2_-treated tumor tissue revealed radiation-induced damages of tumor vessels, endothelial cell swelling, and fibrinoid changes in the wall. The necrotizing effect was associated with the prolongation of γH2AX foci following irradiation with beta rays from ^188^Re-IDA-D-[c(RGDfK)]_2_ (Figure 3E) [30]. To confirm the anti-angiogenic effect in the ^188^Re-IDA-D-[c(RGDfK)]_2_-treated group, we prepared Q-dot 605-labeled RGD peptide for ex vivo scanning in cryosections (Appendix A), and we obtained sufficient evidence of damages in integrin α_v_β_3_ after treatment with ^188^Re-IDA-D-[c(RGDfK)]_2_.

### 3.4. Theranostics for Tumor Angiogenesis

As an integrin-targeted theranostic strategy, ^99m^Tc-IDA-D-[c(RGDfK)]_2_ SPECT was performed in all tested experimental groups (Figure 3) to assess the extent of damage in integrin receptor due to noninvasive beta irradiation of PRRT (Figure 4). Changes in the radionuclide uptake of ^99m^Tc-IDA-D-[c(RGDfK)]_2_ in the tumor region were measured every 7 d. Vehicle- and negative control peptide-treated groups showed a rapid increase in the integrin-mediated uptake levels at 7 d of treatment (13.5 ± 1.8 and 13.1 ± 1.5% IDg^−1^, respectively) and then a slight decrease at 14 d of treatment owing to the rapid enlargement of tumor size. In contrast, SPECT/CT analysis revealed that treatment with ^188^Re-IDA-D-[c(RGDfK)]_2_ significantly suppressed integrin α_v_β_3_ expression and reduced tumor growth in vivo. The tumor volume of the ^188^Re-IDA-D-[c(RGDfK)]_2_-treated group at 14 d of treatment was similar to that of the vehicle-treated group at 7 d of treatment (Figure 4A); however, the percentage IDg^−1^ in the tumor region displayed differences between the two groups (13.5 ± 1.8 and 9.02 ± 0.6% IDg^−1^, respectively; *p* < 0.05).

### 3.5. PRRT Combined with TMZ

To evaluate the combined antitumoral activity of internal radiation and chemotherapy in gliomas, we selected the anticancer drug TMZ (Figure 5), which represents the standard therapy for glioblastoma, although its dose-dependent side effects often limit its use.

TMZ treatment significantly reduced tumor size compared with saline treatment but without any significant dose-dependent differences. Although no dose-dependent responses were observed until day 14 of treatment, histological results showed that high doses of TMZ (5 mgkg^−1^) reduced integrin expression in tumor tissues (Figure 5C,D). The combined use of ^188^Re-IDA-D-[c(RGDfK)]_2_ and TMZ (2 mgkg^−1^) resulted in the best anti-tumor results as confirmed by histological examination that demonstrated a significant reduction in microvessel density (97.9% reduction of positive % area for CD31 expression) and integrin receptors (98.5% reduction of positive % area for integrin α_v_β_3_ expression) (Figure 5C). These results revealed that combined therapy with ^188^Re-IDA-D-[c(RGDfK)]_2_ and TMZ could deplete the angiogenic process and retard glioblastoma progression.

## 4. Discussion

In this study, we demonstrated that ^99m^Tc-IDA-D-[c(RGDfK)]_2_ SPECT could perform the role of diagnostics and assess integrin α_v_β_3_ target density changes during treatment with ^188^Re-IDA-D-[c(RGDfK)]_2_ in U87-MG-bearing mice. ^99m^Tc-IDA-D-[c(RGDfK)]_2_ SPECT might be helpful not only for selecting individuals for RGD peptide-based treatment but also for evaluating any changes in integrin α_v_β_3_ levels and presumably in tumor size after anti-angiogenic treatment. Our preclinical and clinical results suggest that ^99m^Tc-IDA-D-[c(RGDfK)]_2_ could be a novel radiopharmaceutical medical tool.

Based on our previous findings on ^188^Re-IDA-D-[c(RGDfK)]_2_ accumulation in tumor (12.3 ± 1.7% IDg^−1^ at 30 min post-injection) [18], we performed an integrin-targeted radionuclide therapy using ^188^Re-IDA-D-[c(RGDfK)]_2_ in the U87-MG xenograft model. Several previous studies have performed radiotherapy in a subcutaneous mouse xenograft using ^177^Lu, ^90^Yo, or ^67^Cu-labeled RGD peptide and reported their low therapeutic efficacy and requirement in multiple injections of even high doses (37 MBq) [16,31,32]. However, our current findings indicated that ^188^Re-IDA-D-[c(RGDfK)]_2_ effectively suppressed tumor growth, presumably by destroying integrin with Re-188 beta emission. Furthermore, the radioactivity (11.1 MBq) used in the present study was relatively low (0.018 MBqg^−1^) and could be appropriate for clinical application. Nonetheless, multiple injections of ^188^Re-IDA-D-[c(RGDfK)]_2_ were required every 4 d, and thus, further optimization of radiotherapy is needed along with a detailed toxicity test. Moreover, although the antitumor efficacy results found using the subcutaneous tumor model are better than those obtained by [16,31,32], a further criticality of this work could lie in not having used an orthotopic model to be able to explore the BBB crossing ability of our system. However, this aspect will be explored in the future.

Although angiogenesis plays a critical role in tumor growth, it is uncertain whether the PRRT of ^188^Re-IDA-D-[c(RGDfK)]_2_ could be considered a direct cancer therapy. TMZ is an FDA-approved DNA alkylation agent that typically improves survival rates in glioblastoma patients. Since ^188^Re-IDA-D-[c(RGDfK)]_2_ and TMZ have different mechanisms of actions and stoichiometric ratios, we hypothesized that their combination might have a synergistic effect in the treatment of glioma cancer. Reported combination studies show that relatively low doses of the TMZ regime (<7 mgkg^−^^1^, once a week) were effective in the reduction in tumor growth of U87-MG xenografts [33,34]. In our study, we selected the 2 and 5 mgkg^−^^1^ doses for the TMZ single treatment by considering the incidence of adverse effects related to TMZ. Moreover, we performed PRRT combined with the lower one of these two TMZ doses for the purpose of suggesting that a relatively low dose of TMZ could show an inhibition effect of tumor growth in combination therapy.

Glioblastoma growth is closely associated with the formation of new vessels. In the early stages of glioma development, there is no apparent disruption of the BBB; tumor own vasculature has not yet been formed, and the tumor mass is sustained by normal brain vessels. As glioma progresses and aggravates, endothelial cells derived from normal vessels are roughly separated from the vessel main structure and form new angiogenic spots associated with the tumor site. In this context, the importance of our findings about the combined effect of 2 mg TMZ with the radionuclide emerges, since giving the best therapeutic output in the treatment of angiogenic depletion with glioblastoma progress retardation enables keeping the blood–brain barrier intact and thus avoiding progression of the disease [35].

Compared with the relatively inadequate blocking response of cold Re-RGD peptide in integrin receptors (Figure 2), treatment with ^188^Re-IDA-D-[c(RGDfK)]_2_ had a very significant effect on tumor growth in the U87-MG xenograft model, even at relatively small doses (3.7 MBq). Beta irradiation from RGD peptide inhibited tumor growth, presumably by suppressing angiogenesis and destroying integrin α_v_β_3_ in the tumor. In addition, ^188^Re-IDA-D-[c(RGDfK)]_2_ bound to integrin α_v_β_3_ had a crossfire effect that led to DNA damage with subsequent tumor cell death (Figure 3E). Consequently, our current findings indicated that ^188^Re-IDA-D-[c(RGDfK)]_2_ was an effective integrin-targeted radiotherapy.

The most important issue in targeted radiotherapy is the selection of agents that will provide optimal treatment efficacy with minimum undesired side effects due to ineffective exposure. While agent selection is nominally based on tissue type or tumor selectivity, it is impossible to predict its effectiveness in cases of multiple lesions or new tumors at different sites. Therefore, it is desirable to combine similar radiotherapeutic agents and evaluate their behavior for specific localization of a particular tumor target. In the present study, we demonstrated an integrated radiodiagnostic and radiotherapeutic agent set. ^99m^Tc-IDA-D-[c(RGDfK)]_2_ was used to evaluate the possible extent of tumor localization through binding to a tumor-specific integrin α_v_β_3_ target and further assess the change in integrin α_v_β_3_ target density during treatment with ^188^Re-IDA-D-[c(RGDfK)]_2_ (Figure 4A). Our results showed that ^99m^Tc-IDA-D-[c(RGDfK)]_2_ SPECT might be useful not only for selecting individuals for RGD peptide-based treatment but also for evaluating changes in integrin α_v_β_3_ levels and presumably tumor size after radiotherapeutic treatment. While ^188^Re-IDA-D-[c(RGDfK)]_2_ provides the possibility of using integrin-targeted radiotherapy in tumors, there are several limitations to our study. First, we used multiple injections of ^188^Re-IDA-D-[c(RGDfK)]_2_ every 4 d because of the in vivo rapid clearance of RGD peptide. Second, our in vivo efficacy studies did not include any toxicity test, and the safety of radiotherapy was only estimated by the mouse body weight. Although ^188^Re-IDA-D-[c(RGDfK)]_2_ showed a potential PRRT for tumor angiogenesis, special care such as with the kidneys and liver function test panel assay is required to prevent renal and hepatic toxicity. Third, we used only U87-MG glioblastoma cells over a 14 d treatment period. Future studies should be performed in complementary glioma models for a longer period. 

The present study suggested that the combination of PRRT and TMZ might be an effective and synergistic glioblastoma treatment. Internal radiotherapy of tumor angiogenesis combined with chemotherapy and angiogenesis imaging could help to successfully develop new anti-angiogenesis drugs. Further studies examining the efficacy of combined therapy in other glioma models are needed to confirm whether ^188^Re-IDA-D-[c(RGDfK)]_2_ can improve the treatment of GBM.

## 5. Conclusions

In conclusion, the results of biodistribution, pharmacokinetics, in vivo SPECT imaging, and anti-angiogenic radiotherapy efficacy studies suggest that ^99m^Tc- and ^188^Re-IDA-D-[c(RGDfK)]_2_ are promising theranostic tools in the field of tumor-induced angiogenesis. Combined therapy with PRRT and TMZ showed more cytotoxic effects than monotherapy. Overall, ^188^Re-IDA-D-[c(RGDfK)]_2_ might be a valuable and innovative radiotherapeutic agent for the treatment of GBM.

## Figures and Tables

**Figure 1 cancers-13-05029-f001:**
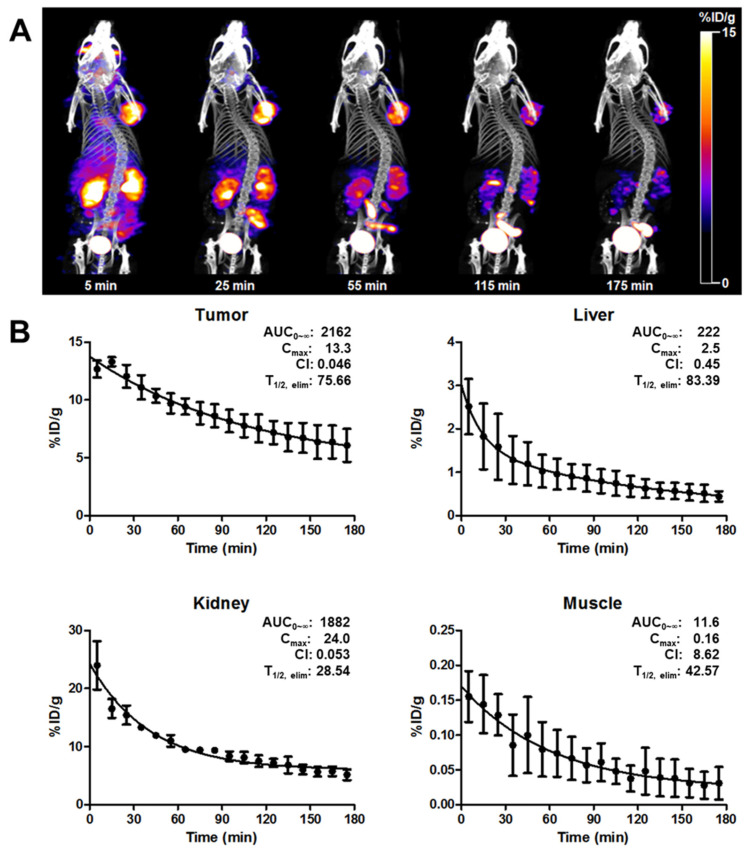
In vivo characteristics of ^99m^Tc-IDA-D-[c(RGDfK)]_2_. Representative serial SPECT/CT images of ^99m^Tc-IDA-D-[c(RGDfK)]_2_ in the U87-MG xenograft (**A**). ROI-derived radionuclide uptake of tumor, liver, kidneys, and muscle post-injection of ^99m^Tc-IDA-D-[c(RGDfK)]_2_ (**B**). Color bars indicate the range of radionuclide uptake as %IDg^−1^. Results are given as the means ± SD (*n* = 3). AUC_0-__∞_: the area under the concentration-time curve (%IDg^−1^∙min), C_max_: the maximum concentrations (%IDg^−1^), Cl: the clearance rate (mLmin^−1^); T_1/2 elim_: the elimination half-life (min).

**Figure 2 cancers-13-05029-f002:**
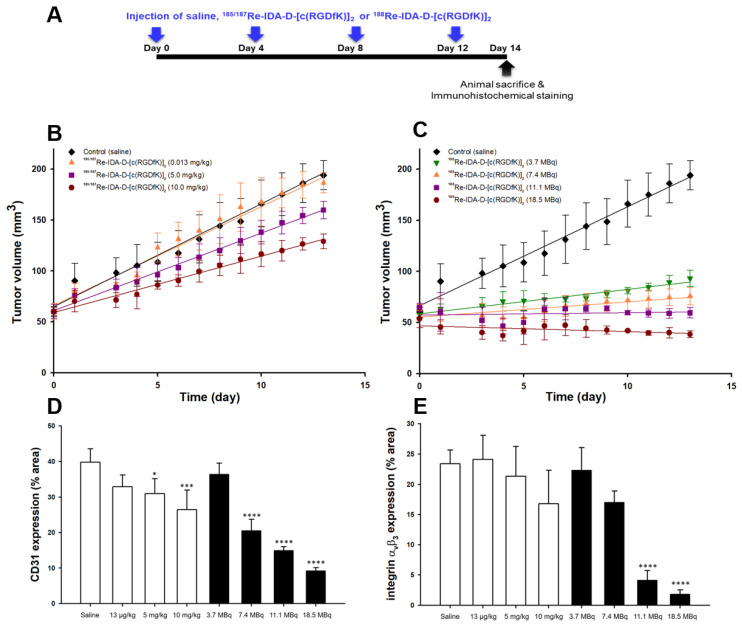
Effects of integrin targeted blocking and radiotherapy on the growth of U87-MG xenografts. (**A**) Schematic protocol of the treatment. (**B**) The authentic “cold” Re-peptide (^185/187^Re-IDA-D-[c(RGDfK)]_2_) dose response for tumor volume. (**C**) Radiotherapy (^188^Re-IDA-D-[c(RGDfK)]_2_) dose response for tumor volume. Four mice were used for each time point. (**D**) Microvessel and (**E**) integrin α_v_β_3_ positive % area of tumors treated with either saline, ^185/187^Re-IDA-D-[c(RGDfK)]_2_ (0.013, 5, and 10 mgkg^−1^), or ^188^Re-IDA-D-[c(RGDfK)]_2_ (3.7, 7.4, 11.1, and 18.5 MBq) after 2 weeks. Expression levels of microvessel and integrin α_v_β_3_ were evaluated in all tested experimental groups. Error bars represent means ± SD. Data were analyzed by ANOVA test and post hoc test (each group versus control group, * *p* < 0.05, *** *p* < 0.001, **** *p* < 0.0001).

**Figure 3 cancers-13-05029-f003:**
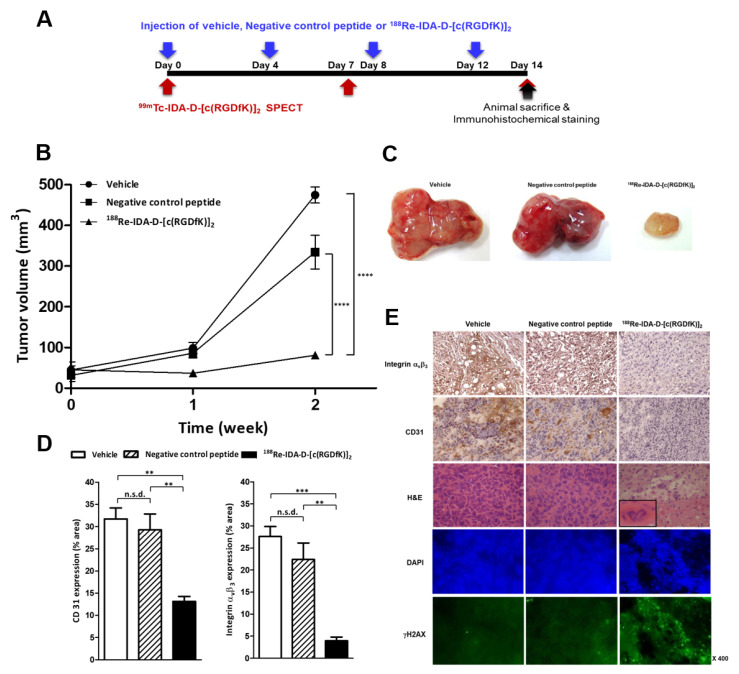
Anti-angiogenic effects of ^188^Re-IDA-D-[c(RGDfK)]_2_ (11.1 MBq) in U87-MG xenografts. Tumor-bearing mice were treated with either vehicle, negative control peptide (^188^Re-IDA-D-[c(RADfK)]_2_), or ^188^Re-IDA-D-[c(RGDfK)]_2_ for 2 weeks (*n* = 4 for each). (**A**) Schematic protocol of the treatment. (**B**) Tumor volume at each time point. (**C**) Representative macroscopic appearance of the dissected tumors in three groups. (**D**) Microvessel and integrin α_v_β_3_ positive % area of tumors treated in all tested experimental groups after 2 weeks. (**E**) Immunohistochemical staining with anti-α_v_β_3_, anti-CD31, and H&E and immunofluorescence images with DAPI and γH2AX for DNA damage in the tumor sections. Data were analyzed by ANOVA test and post hoc test (Bonferroni correction, ** *p* < 0.01, *** *p* < 0.001, **** *p* < 0.0001). Abbreviation: NSD means no significant difference.

**Figure 4 cancers-13-05029-f004:**
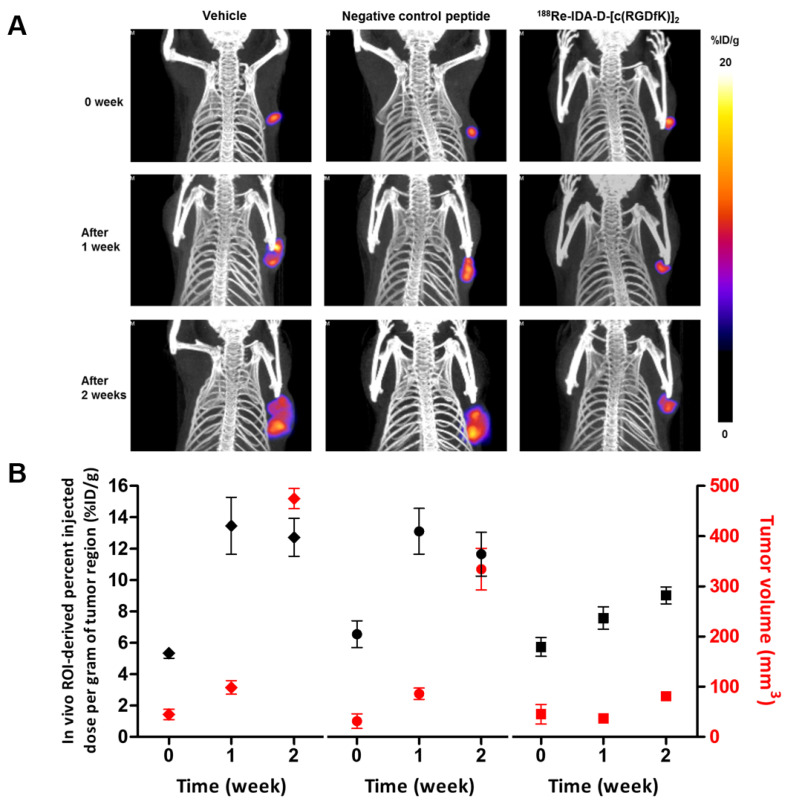
Differences in the uptake of ^99m^Tc-IDA-D-[c(RGDfK)]_2_ in images in grouped mice as %IDg^−1^ indicate considerable anti-angiogenic response. (**A**) ^99m^Tc-IDA-D-[c(RGDfK)]_2_ SPECT imaging of integrin α_v_β_3_ in the U87-MG xenografts before and after administration of the vehicle (closed lozenge), negative control peptide (^188^Re-IDA-D-[c(RADfK)]_2_, closed circle) or ^188^Re-IDA-D-[c(RGDfK)]_2_ (closed square) treatment. (**B**) Measurement of %IDg^−1^ of ^99m^Tc-IDA-D-[c(RGDfK)]_2_ in the tumor region (black color, left y-axis) and tumor volume at each time (red color, right y-axis) from mice treated with vehicle, negative control peptide or ^188^Re-IDA-D-[c(RGDfK)]_2_.

**Figure 5 cancers-13-05029-f005:**
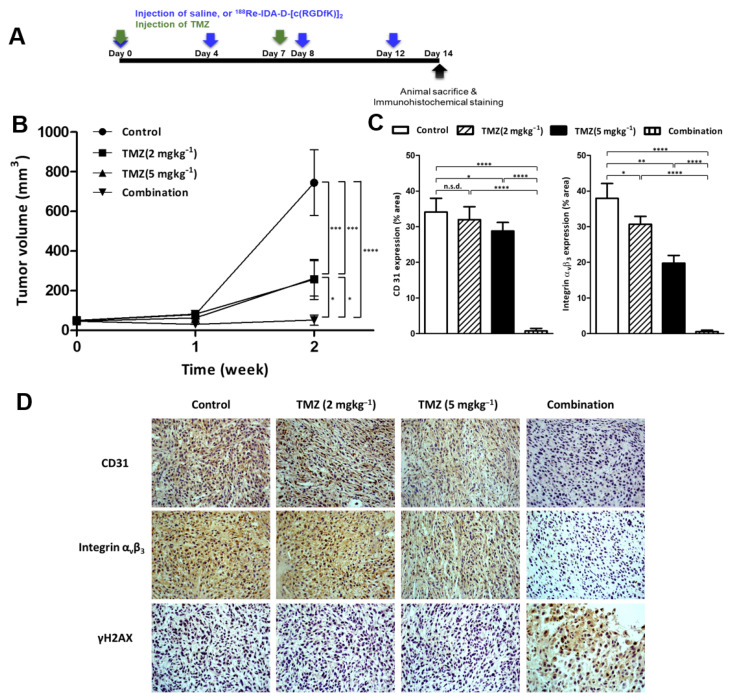
Therapeutic effect of ^188^Re-IDA-D-[c(RGDfK)]_2_ combined with TMZ in the U87-MG xenografts. Tumor-bearing mice were treated with either control (saline), TMZ (2 or 5 mgkg^−1^), or a combination (11.1 MBq of ^188^Re-IDA-D-[c(RGDfK)]_2_ with 2 mgkg^−1^ of TMZ) for 2 weeks (*n* = 4 for each). (**A**) Schematic protocol of the treatment. (**B**) Tumor volume at each time point. (**C**) Microvessel and integrin α_v_β_3_ positive % area of tumors treated in all tested experimental groups after 2 weeks. (**D**) Immunohistochemical staining of tumor slices with anti-α_v_β_3_, anti-CD31, and γH2AX in tumor sections of all tested experimental groups after 2 weeks. Data were analyzed by ANOVA test and post hoc test (Bonferroni correction, * *p* < 0.05, ** *p* < 0.01, *** *p* < 0.001, **** *p* < 0.0001). Abbreviation: NSD means no significant difference.

## Data Availability

All data are contained within the article or the Appendix A.

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
