# Peer review of "Effect of Peptide Receptor Radionuclide Therapy in Combination with Temozolomide against Tumor Angiogenesis in a Glioblastoma Model"

_cancers, 2021, doi:10.3390/cancers13195029_

Round 1

Reviewer 1 Report

The manuscript submitted by Lee SH et al. reports the evaluation of combining 188Re-IDA-D-[c(RGDfK)]2 and temozolomide for the treatment of glioblastoma. Temozolomide is an approved alkylating agent for use to treat glioblastoma and 188Re-IDA-D-[c(RGDfK)]2 is a potent binder of integrin αvβ3 which is highly expressed by glioblastoma. Thorough studies were conducted in U87-MG tumor-bearing mice, and the results showed that combination of 188Re-IDA-D-[c(RGDfK)]2 and temozolomide is more efficacious than 188Re-IDA-D-[c(RGDfK)]2 or temozolomide alone. In addition, they also showed that 99mTc-IDA-D-[c(RGDfK)]2 can be used as diagnostic imaging agent to evaluate the treatment response, and can be coupled with 188Re-IDA-D-[c(RGDfK)]2 as a theranostic pair for the management of glioblastoma. The reported research is novel, the data are promising, the conclusions are supported by the presented data, and this manuscript is well written. Therefore, this manuscript can be accepted for publication after the following suggested minor changes have been addressed:

  • Line 287: “The number of CD-31 positive microvessels were decreased to 22% at 5 mg kg-1 and 33% at 10 mg kg-1 in the 185/187Re-IDA-D-[c(RGDfK)]2-treated group.” Please double check the numbers as these are different from the data presented in Fig 2C.
  • Fig 2 legend: Please specify what are presented in (C) and (D).
  • Fig 5 legend: Please specify what are presented in (B).
  • Line 384: “... that demonstrated a significant reduction in microvessel density (98.5% reduction of positive % area for integrin αvβ3 expression) and integrin receptors (97.9% reduction of positive % area for CD31 expression) (Figure 5C).” should be “... that demonstrated a significant reduction in microvessel density (97.9% reduction of positive % area for CD31 expression) and integrin receptors (98.5% reduction of positive % area for integrin αvβ3 expression) (Figure 5C).”.
  • Last paragraph in Discussion Section: add a limitation for using a subcutaneous tumor model rather than an orthotopic model as 188Re-IDA-D-[c(RGDfK)]2 might not cross blood-brain barrier well.  

Author Response

We thank the reviewer for her/his comments. Please see the attachment

Reviewer 2 Report

The authors describe the novel approach of the combinations of radiotherapeutics with Temozolomide in the Glioblastoma treatments which could be helpful in next generation treatment regime. To address the potential limitations and contraindications of such approach, there are some question needs to address:

  1. In the result section 3.1, To establish the efficacy and pharmacokinetics of radionuclide Tc-RGD compound the authors should describe the liver toxicity and renal toxicity assessment by providing the kidney and liver function test panel assay and its compatibility with human subjects.
  2. The results shows that the combined effect of 2mgTMZ with the radionuclide gives the best therapeutic output in the treatment of angiogenic depletion with glioblastoma progress retardation and the author should describe that the encouraging findings can keep the blood-brain barrier mechanism intact.

Author Response

We thank the reviewer for her/his comments. Please see the attachment.

Reviewer 3 Report

I thank the authors for this interesting paper analyzing a new therapeutic option with a combination therapy in GBM. The results indicated that integrin-targeted PRRT  and TMZ combined therapy might be a new medical tool for the effective treatment of glioblastoma, however the presentation of the methods and results are not clear enough.

The quality of immunohistochemical stainings is partly not sufficient. Figure legends need to be much more detailed. 

1) The two first sentences in introduction need a reference.

2) methods: -2.3.: the experimental set-up needs to be presented much better with a schematic figure. Was the treatment initiation planned for a specific date after tumor inoculation oder was it volume dependent? Then please state the exact volume you aimed to start the treatment. As it is now, you reported the volumes with SD at the beginning. Please state also at which day after the tumor inoculation these volumina was reached in the text. 

-the animal experiments should be reported using arrive guideline including sex, weight, exact description of mouse strain, number of animals for EACH group. Also adverse effects and weight curves should be presented for the tolerability of the therapeutic arms.

2.5.: please demonstrate the experimental set-up with a schematic figure

2.10: the statistical tests need to be reported not only the softwares !!!for osmntance for three groups anova with bonf. correction would be needed etc. was spss or sigmaplot used? both are stated.

Figure 3a: abosulte volumes should also be compared to each other, not only relative tumor growth, as the initial tumor size were comparable amongst all groups (the difference can also be tested.) In the figure legens, it mus be stated per diagram which test was performed NOT the program used...also the number of animals per diagramm MUST be stated.

Figure 2 legend: for c and d as letters are missing. what is 100% here?

figure 3e: dapi seems to be "not well done" , the cell nuclei are blurr, low quality. DNA damage staining is also not very specific. For all the stainings, the day of the tumor sacrifice for the assessment should be written. 

figure 4: in the methods is says the tumor cells were implanted in the right shoulder, the image shows other location. also here, please presenr also absolute volumes in comparison to each other

Figure 5: here also absolute tumor volumes. Then I do not understand the calculation cd 31 area in %, what is 100% Legend für B is missing! this is also the case in other figures comparing immunothisto. stainings. 

The discussion is not addressing important issues such as choice of TMZ regime used here (one a week?, why?). Then, the major issue is the blood brain barrier for GBM as the authors used a subcut. tumor model. This issue has to be addressed in the discussion with a future perspective.

Author Response

(The authors gave the same response as above.)

Round 2

Reviewer 3 Report

see the document please
